# Towards unlocking the mystery of adversarial fragility of neural networks

## Abstract

In this paper, we study the adversarial robustness of deep neural networks for classification tasks. We look at the smallest magnitude of possible additive perturbations that can change the output of a classification algorithm. We provide a matrix-theoretic explanation of the adversarial fragility of deep neural network for classification. In particular, our theoretical results show that neural network's adversarial robustness can degrade as the input dimension $d$ increases. Analytically we show that neural networks' adversarial robustness can be only $1/\sqrt{d}$ of the best possible adversarial robustness. Our matrix-theoretic explanation is consistent with an earlier information-theoretic feature-compression-based explanation for the adversarial fragility of neural networks.

## 1 Introduction

Deep learning or neural network based classifiers are known to offer high classification accuracy in many classification tasks. However, it is also observed that deep learning based classifiers often suffer from adversarial fragility and have low robustness under adversarial perturbations Szegedy et al. (2014); Goodfellow et al. (2014). For example, when a small amount of adversarial noise is added to the signal input of a deep learning classifier, its output can dramatically change from an accurate label to an inaccurate label, even though the input signal is barely changed according to human perceptions. The reason why the deep learning classifier is often fragile has remained a mystery, even though there have been various theories explaining this phenomenon, see e.g. Akhtar & Mian (2018); Yuan et al. (2017); Huang et al. (2018); Wu et al. (2024); Wang et al. (2023) for surveys.

These studies, however, have not yet resulted in a consensus on the important question: a theoretical explanation for adversarial fragility. Instead, we currently have multiple competing theoretical explanations, which include (a) quasi-linearity/smoothness of the decision function in AI classifiers Goodfellow et al. (2014); Li & Spratling (2023); Kanai et al. (2023); Eustratiadis et al. (2022), (b) high curvature of the decision boundary Fawzi et al. (2016); Reza et al. (2023); Singla et al. (2021), (c) closeness of the classification boundary to the data sub-manifold Tanay & Griffin (2016); Zeng et al. (2023); Xu et al. (2022), and (d) information-theoretic feature compression hypothesis Xie et al. (2019). In Ilyas et al. (2019), the authors argued that the adversarial fragility of neural network possibly came from the neural network utilizing non-robust features for classification. However, there are recent works, for example Li et al. (2023), which show that non-robust features might not be able to fully explain the adversarial fragility of neural network based classifiers.

Besides these works, there are results trying to use high dimensional statistical analysis tools to theoretically understand the adversarial robustness of classification models. An asymptotically exact formula given in Hassani & Javanmard (2022) shows that higher overparametrization leads to a worse robust generalization error for the adversarially-trained models. The performance of high-dimensional linear models is studied in Donhauser et al. (2021) and Javanmard et al. (2020) which showed that the robust generalization error of adversarially-trained models becomes worse as the models become more overparameterized. The analysis in Taheri et al. (2021) is for the adversarially-trained linear model in the high-dimensional regime where the dimension of data grows with the size of the training data-set at a constant ratio for binary classification. In Taheri et al. (2021), the authors precisely analyzed the performance of adversarial training with $\ell_2$ and $\ell_\infty$-norm bounded perturbations in binary classification for Gaussian mixture and generalized linear models. It was shown in Tsipras et al. (2019) that there exists a trade-off between the standard accuracy of a model and its robustness to adversarial perturbations. It is also observed that using more data can improve this trade-off

Carmon et al. (2022); Min et al. (2021); Najafi et al. (2019); Raghunathan et al. (2019); Rebuffi et al. (2021).

Despite these efforts, there is still not clear consensus or theoretical understanding of the fundamental reason for the adversarial fragility of neural network based classifiers Li et al. (2023). It might be tempting to explain the adversarial fragility of neural network based classifiers purely as the gap between the average-case performance (the performance of the classifier under random average-case noise) and the worst-case performance (the performance of the classifier under well-crafted worst-case perturbation), for example through the linearity of the model Goodfellow et al. (2014). However, we argue that this average-case-versus-worst-case gap cannot explain the dramatic fragility of deep learning based classifiers. Firstly, it is common that there is a gap between average-case and worst-case performance: it exists for almost every classifier (even including theoretically optimal classifiers), and is not particularly tied to neural network based classifiers. Secondly, we can show that there exists good-performing classifiers whose worst-case performance are provably orders of dimension better than the worst-case performance of deep learning based classifiers. So there are deeper reasons for the adversarial fragility of neural network based classifiers than just the worst-case-versus-average-case degradation.

In this paper, we study the adversarial robustness of deep neural networks for classification tasks from a different perspective than the current literature. We focus on comparing the worst-case performance of neural network based classifiers and optimal classifiers. We look at the smallest magnitude of possible additive perturbations that can change the output of the classification algorithm. We provide a matrix-theoretic explanation of the adversarial fragility of deep neural network. In particular, our theoretical results show that neural network's adversarial robustness can degrade as the input dimension $d$ increases. Analytically we show that neural networks' adversarial robustness can be only $1/\sqrt{d}$ of the best possible adversarial robustness.

| **This Paper's Comparison** | Optimal classifier | Neural network based classifier |
|---|---|---|
| Worst-case performance | ✓ | ✓ |

In particular, in this paper, through concrete classification examples and matrix-theoretic derivations, we show that the adversarial fragility of neural network based classifiers comes from the fact that very often neural network only uses a subset (or compressed features) of all the features to perform the classification tasks. Thus in adversarial attacks, one just needs to add perturbations to change the small subsets of features used by the neural networks. This conclusion from matrix-theoretic analysis is consistent with the earlier information-theoretic feature-compression-based hypothesis that neural network based classifier's fragility comes from its utilizing compressed features for final classification decisions Xie et al. (2019). Different from Xie et al. (2019) which gave a higher-level explanation based on the feature compression hypothesis and high-dimensional geometric analysis, this paper gives the analysis of adversarial fragility building on concrete neural network architectures and classification examples. Our results are derived for linear and non-linear, for two-layer and general multiple-layer neural networks with different assumptions on network weights, and for different classification tasks. As a byproduct, we developed a characterization of the distribution of the QR decomposition of the products of random Gaussian matrices in Lemma 3.

## 2 PROBLEM STATEMENT

In this section, we review basic notations and architectures for deep learning based classifiers.

We will denote the $\ell_2$ norm of an vector $\mathbf{x} \in \mathbb{R}^n$ by $\|\mathbf{x}\|$ or $\|\mathbf{x}\|_2 = \sqrt{\sum_{i=1}^{n} |\mathbf{x}_i|^2}$. Let a neural network based classifier $G(\cdot) : \mathbb{R}^d \to \mathbb{R}^k$ be implemented through a $l$-layer neural network which has $l - 1$ hidden layers and has $l + 1$ columns of neurons (including the neurons at the input layer and output layer). We denote the number of neurons at the inputs of layers 1, 2, ..., and $l$ as $n_1$, $n_2$, ...., and $n_l$ respectively. At the output of the output layer, the number of neurons is $n_{l+1} = k$, where $k$ is the number of classes.

We define the bias terms in each layer as $\boldsymbol{\delta_1} \in \mathbb{R}^{n_2}, \boldsymbol{\delta_2} \in \mathbb{R}^{n_3}, \cdots, \boldsymbol{\delta_{l-1}} \in \mathbb{R}^{n_l}, \boldsymbol{\delta_l} \in \mathbb{R}^{n_{l+1}}$, and the weight matrices $H_i$ for the $i$-th layer is of dimension $\mathbb{R}^{n_{i+1} \times n_i}$.

The element-wise activation functions in each layer are denoted by $\sigma(\cdot)$, and some commonly used activation functions include ReLU and leaky ReLU. So the output $\mathbf{y}$ when the input is $\mathbf{x}$ is given by $\mathbf{y} = G(\mathbf{x}) = \sigma(H_l \sigma(H_{l-1} \cdots \sigma(H_1 \mathbf{x} + \boldsymbol{\delta_1}) \cdots + \boldsymbol{\delta_{l-1}}) + \boldsymbol{\delta_l})$.

## 3 FEATURE COMPRESSION LEADS TO SIGNIFICANT DEGRADATION IN ADVERSARIAL ROBUSTNESS

In this section, we start presenting the main results of this paper. In particular, we first give theoretical analysis of linear neural network based classifiers' adversarial robustness, and show that the worst-case performance of neural network based classifiers can be orders of magnitude worse than the worst-case performance of optimal classifiers. We then generalize our results to analyze the worst-case performance of non-linear neural network based classifiers for classification tasks with more complicatedly-distributed data.

**Theorem 1.** *Consider $d$ training data points $(\mathbf{x}_i, i)$, where $i = 1, 2, \cdots, d$, each $\mathbf{x}_i$ is a $d$-dimensional vector with each of its elements following the standard Gaussian distribution $\mathcal{N}(0, 1)$, and each $i$ is a distinct label. Consider a two-layer (will be extended to multiple layers in later theorems) neural network whose hidden layer's output is $\mathbf{z} = \sigma(H_1 \mathbf{x} + \boldsymbol{\delta_1})$, where $H_1 \in \mathbb{R}^{m \times d}$, $\mathbf{z} \in \mathbb{R}^{m \times 1}$, and $m$ is the number of hidden layer neurons.*

*For each class $i$, suppose that the output for that class at the output layer of the neural network is given by*

$$f_i(\mathbf{x}) = \mathbf{w}_i^T \sigma(H_1 \mathbf{x} + \boldsymbol{\delta_1}),$$

*where $\mathbf{w}_i \in \mathbb{R}^{m \times 1}$. By the softmax function, the probability for class $i$ is given by $o_i = \frac{e^{f_i}}{\sum_i^k e^{f_i}}$.*

*To simplify our analysis, suppose that the hidden layer's activation function is identity (which will be extended to general functions in Theorem 7), and that $H_1$ is a matrix with orthogonal columns satisfying $H_1^T H_1 = I_{m \times m}$ (which will be extended to general $H_1$ in Theorem 4).*

*For each class $i$, suppose that the neural network satisfies*

$$f_j(\mathbf{x}_i) = \begin{cases} 1, \text{if } j = i, \\ 0, \text{if } j \neq i. \end{cases} \tag{1}$$

*Then we have:*

- *with high probability, for every $\epsilon > 0$, the smallest distance between any two data points is*

$$\min_{i \neq j, \ i=1,2,\ldots,d, \ j=1,2,\ldots,d} \|\mathbf{x}_i - \mathbf{x}_j\|_2 \geq (1 - \epsilon)\sqrt{2d}.$$

  *For each class $i$, one would need to add a perturbation $\mathbf{e}$ of size $\mathbf{e} \geq \frac{(1-\epsilon)\sqrt{2d}}{2}$ to change the classification decision if the minimum-distance classification rule is used.*

- *For each $i$, with high probability, one can add a perturbation $\mathbf{e}$ of size $\|\mathbf{e}\|_2 \leq C$ such that the classification result of the neural network is changed, namely*

$$f_j(\mathbf{x}_i + \mathbf{e}) > f_i(\mathbf{x}_i + \mathbf{e})$$

  *for a certain $j \neq i$, where $C$ is a constant independent of $d$ and $m$.*

*Proof.* To prove the first claim, we need the following lemma (proof provided in the appendix).

**Lemma 2.** *Suppose that $Z_1$, $Z_2$, ... and $Z_d$ are i.i.d. random variables following the standard Gaussian distribution $\mathcal{N}(0, 1)$. Let $\alpha$ be a constant smaller than 1. Then the probability that $\sum_{i=1}^d Z_i^2 \leq \alpha d$ is at most $\left(\alpha(e^{1-\alpha})\right)^{\frac{d}{2}}$. Moreover, as $\alpha \to 0$, the natural logarithm of this probability divided by $d$ goes to negative infinity.*

We consider each pair of $\mathbf{x}_i$ and $\mathbf{x}_j$. Then $\mathbf{x}_i - \mathbf{x}_j$ will be a $d$-dimensional vector with elements as independent zero-mean Gaussian random variables with variance 2. So by Lemma 2, we know with high probability that the distance between $\mathbf{x}_i$ and $\mathbf{x}_j$ will be at least $(1 - \epsilon)\sqrt{2d}$. By taking the union bound over $\binom{d}{2}$ pairs of vectors, we have proved the first claim.

We let $X = [\mathbf{x}_1, \mathbf{x}_2, ..., x_d]$ be a $\mathbb{R}^{d \times d}$ matrix with its columns as $\mathbf{x}_i$'s. Without loss of generality, we assume that the ground-truth signal is $\mathbf{x}_d$ corresponding to label $d$.

Then we consider the QR decomposition of $H_1 X$,

$$H_1 X = Q_1 R,$$

where $Q_1 \in \mathbb{R}^{m \times d}$ satisfies $Q_1^T \times Q_1 = I_{d \times d}$, and $\mathbb{R}^{m \times d}$ is an upper-triangular matrix. We further consider the QR decomposition of $X$ as

$$X = Q_2 R,$$

where $Q_2 \in \mathbb{R}^{d \times d}$ and $Q_1 = H_1 Q_2$. Note that the two matrices $R$ above are the same matrix due to orthogonality of $H_1$.

Because of condition (1), the weight matrix $H_2$ between the hidden layer and the output layer is

$$H_2 = R^{-1} Q_2^T H_1^T = R^{-1} Q_1^T.$$

So when the input to the neural network is

$$\mathbf{x}_d = Q_2 \times [R_{1,d}, R_{2,d}, R_{3,d} \ldots R_{d-2,d} R_{d-1,d} R_{d,d}]^T,$$

the $d$ outputs at the $d$ output neurons are

$$\mathbf{y} = H_2 H_1 \mathbf{x}_d = H_2 H_1 Q_2 \begin{bmatrix} R_{1,d} \\ R_{2,d} \\ R_{3,d} \\ \ldots \\ R_{d-2,d} \\ R_{d-1,d} \\ R_{d,d} \end{bmatrix} = \begin{bmatrix} 0 \\ 0 \\ 0 \\ \ldots \\ 0 \\ 0 \\ 1 \end{bmatrix}.$$

We let $\mathbf{e} = Q_2 \times \mathbf{e}_{basis}$, where $\mathbf{e}_{basis} = (0, 0, ..., 0, R_{d-1,d-1} - R_{d-1,d}, -R_{d,d})^T$. We claim that under such a perturbation $\mathbf{e}$, the input will be $\mathbf{x}_d + \mathbf{e}$ and we will have $f_d(\mathbf{x}_d + \mathbf{e}) = 0$ and $f_{d-1}(\mathbf{x}_d + \mathbf{e}) = 1$. In fact, when the input is $\mathbf{x}_d + \mathbf{e}$, the output at the $d$ output neurons is given by

$$\mathbf{y} = R^{-1} \left( \begin{bmatrix} R_{1,d} \\ R_{2,d} \\ R_{3,d} \\ \ldots \\ R_{d-2,d} \\ R_{d-1,d} \\ R_{d,d} \end{bmatrix} + \begin{bmatrix} 0 \\ 0 \\ 0 \\ \ldots \\ 0 \\ R_{d-1,d-1} - R_{d-1,d} \\ -R_{d,d} \end{bmatrix} \right).$$

We focus our attention on the outputs of the last two output neurons, and show that the classification result will be changed to an incorrect one under the current perturbation. To see this, we first notice that the inverse of $R$ is an upper triangular matrix given by

$$\begin{bmatrix} * & * & * & \ldots & * & * & * \\ 0 & * & * & \ldots & * & * & * \\ 0 & 0 & * & \ldots & * & * & * \\ & & & \ldots & & & \\ 0 & 0 & 0 & \ldots & 0 & \frac{1}{R_{d-1,d-1}} & -\frac{R_{d-1,d}}{R_{d-1,d-1} \cdot R_{d,d}} \\ 0 & 0 & 0 & \ldots & 0 & 0 & \frac{1}{R_{d,d}} \end{bmatrix},$$

where we only explicitly write down the last two rows.

We know that $\mathbf{x}_d = Q_2 R_{:,d}$, where $R_{:,d}$ is the last column of $R$. Then $(f_{d-1}(\mathbf{x}_d + \mathbf{e}), f_d(\mathbf{x}_d + \mathbf{e}))^T$ is equal to

$$\begin{bmatrix} 0 & 0 & 0 & \ldots & \frac{1}{R_{d-1,d-1}} & -\frac{R_{d-1,d}}{R_{d-1,d-1} \cdot R_{d,d}} \\ 0 & 0 & 0 & \ldots & 0 & \frac{1}{R_{d,d}} \end{bmatrix} \begin{bmatrix} 0 + R_{1,d} \\ 0 + R_{2,d} \\ 0 + R_{3,d} \\ \ldots \\ 0 + R_{d-2,d} \\ (R_{d-1,d-1} - R_{d-1,d}) + R_{d-1,d} \\ (-R_{d,d}) + R_{d,d} \end{bmatrix} \tag{2}$$

$$= \begin{bmatrix} \frac{R_{d-1,d-1}}{R_{d-1,d-1}} + \frac{0}{R_{d-1,d-1} \cdot R_{d,d}} \\ 0 \end{bmatrix} = \begin{bmatrix} 1 \\ 0 \end{bmatrix}. \tag{3}$$

The magnitude of this perturbation is

$$\|\mathbf{e}\|_2 = \|Q_2 \mathbf{e}_{basis}\|_2 = \sqrt{(R_{d-1,d-1} - R_{d-1,d})^2 + (-R_{d,d})^2} \leq |R_{d-1,d-1}| + |R_{d-1,d}| + |R_{d,d}|.$$

By random matrix theory Hassibi & Vikalo (2005); Xu et al. (2004), $R_{d,d}$ is the absolute value of a random variable following the standard Gaussian distribution $\mathcal{N}(0,1)$. Moreover, $R_{d-1,d-1}$ is the square root of a random variable following the chi-squared distribution of degree 2; and $R_{d-1,d}$ is a standard normal random variable. Thus, there exists a constant $C$ such that, with high probability, under an error $\mathbf{e}$ with $\|\mathbf{e}\|_2 \leq C$, the predicted label of the neural network will be changed. □

**Remarks:** Note that $\mathbf{x}_d = \sum_{i=1}^{d}(Q_2)_{:,i}R_{i,d}$, where $(Q_2)_{:,i}$ is the $i$-th column of $Q_2$. However, to attack this classifier, we only need to attack the features in the direction $(Q_2)_{:,d}$ which the classifier uses for making decisions.

Now we go beyond 2-layer neural networks, and moreover, consider the general case where $H_1$, $H_2$, $H_3$, ..., and $H_{l-1}$ are general matrices whose elements are i.i.d. standard normal random variables, instead of being orthonormal matrices. For these general matrices, we have the following novel characterization of the $QR$ decomposition of their products (see the proof in the appendix).

**Lemma 3.** *Let $H = H_{l-1} \cdots H_2 H_1$, where each $H_i$ ($1 \leq i \leq l-1$) is an $n_{i+1} \times n_i$ matrix composed of i.i.d. standard zero-mean unit-variance Gaussian random variables, and $H_i$'s are jointly independent. Here without loss of generality, we assume that for every $i$, $n_{i+1} \geq n_i$.*

*We let $R_1$, $R_2$, ...., and $R_{l-1}$ be $l-1$ independent upper triangular matrices of dimension $n_1 \times n_1$ with random elements in the upper-triangular sections. In particular, for each $R_i$, $1 \leq i \leq l-1$, its off-diagonal elements in the strictly upper triangular section are i.i.d. standard Gaussian random variables following distribution $\mathcal{N}(0,1)$; its diagonal element in the $j$-th row is the square root of a random variable following the chi-squared distribution of degree $n_{i+1} - j + 1$, where $1 \leq j \leq n_1$.*

*Suppose that we perform QR decomposition on $H$, namely $H = QR$, where $R$ is of dimension $n_1 \times n_1$. Then $R$ follows the same probability distribution as $R_{l-1}R_{l-2} \cdots R_2 R_1$, namely the product of $R_1$, $R_2$, ..., and $R_{l-1}$.*

Now we are ready to extend Theorem 1 to more general multiple-layer neural network with general weights.

**Theorem 4.** *Consider $d$ data points $(\mathbf{x}_i, i)$, where $i = 1, 2, \cdots, d$, each $\mathbf{x}_i$ is a $d$-dimensional vector with each of its elements following the standard Gaussian distribution $\mathcal{N}(0,1)$, and each $i$ is a distinct label. Consider a multiple-layer linear neural network whose hidden layers' output is*

$$\mathbf{z} = H_{l-1}...H_1\mathbf{x}, \tag{4}$$

*where $H_i \in \mathbb{R}^{n_{i+1} \times n_i}$, and $n_1 = d$. For each class $i$, suppose that the output for that class at the output layer of the neural network is given by $f_i(\mathbf{x}) = \mathbf{w}_i^T \mathbf{z}$, where $\mathbf{w}_i \in \mathbb{R}^{n_{l+1} \times 1}$. By the softmax function, the probability for class $i$ is given by $o_i = \frac{e^{f_i}}{\sum_i^k e^{f_i}}$. For each class $i$, suppose that the neural network satisfies*

$$f_j(\mathbf{x}_i) = \begin{cases} 1, \text{if } j = i, \\ 0, \text{if } j \neq i. \end{cases} \tag{5}$$

*Then we have:*

- *with high probability, for every $\epsilon > 0$, the smallest distance between any two data points is*

$$\min_{i \neq j, \ i=1,2,...,d, \ j=1,2,...,d} \|\mathbf{x}_i - \mathbf{x}_j\|_2 \geq (1-\epsilon)\sqrt{2d}.$$

*For each class $i$, one would need to add a perturbation $\mathbf{e}$ of size $\mathbf{e} \geq \frac{(1-\epsilon)\sqrt{2d}}{2}$ to change the classification decision if the minimum-distance classification rule is used.*

- *For each class $i$, with high probability, one can add a perturbation $\mathbf{e}$ of size $\|\mathbf{e}\|_2 \leq C$ such that the classification result of the neural network is changed, namely*

$$f_j(\mathbf{x}_i + \mathbf{e}) > f_i(\mathbf{x}_i + \mathbf{e})$$

*for a certain $j \neq i$, where $C$ is a constant independent of $d$.*

*Proof.* The first part is proved in Theorem 1. For the second part, we use Lemma 3. From the proof of Theorem 1, we know we just need to add perturbation with magnitude at most $|R_{d-1,d-1}| + |R_{d-1,d}| + |R_{d,d}|$, where $R$ the upper triangular matrix resulting from the QR decomposition of $H_{l-1}...H_1$. Moreover, by Lemma 3,

$$|R_{d-1,d-1}| + |R_{d-1,d}| + |R_{d,d}| \leq \|R_{l-1}\|_{1B}...\|R_1\|_{1B},$$

where $\|R_i\|_{1B}$ is the sum of the absolute values of elements in the bottom $2 \times 2$ submatrix of $R_i$. Because with high probability, $\|R_{l-1}\|_{1B}, ..., \|R_1\|_{1B}$ will all be bounded by a constant $D$ at the same time, we can find a perturbation of size bounded by a constant $D^l$ such that changes the output decision of the neural network classifier. □

So far we have assumed for multiple-layer neural network, the following condition holds: for each class $i$, suppose that the neural network satisfies

$$f_j(\mathbf{x}_i) = \begin{cases} 1, \text{if } j = i, \\ 0, \text{if } j \neq i. \end{cases} \tag{6}$$

This condition facilitates characterizing the adversarial robustness of neural networks via random-matrix-theoretic analysis of the QR decomposition of a Gaussian matrix. For general last layer's weights which do not necessarily satisfy this condition, we have the following results.

**Theorem 5.** *Consider a multi-layer linear neural network for the classification problem in Theorem 1. Suppose that the input signal $\mathbf{x}$ corresponds to a ground-truth class $i$. Let us consider an attack target class $j \neq i$. Let the last layer's weight vectors for class $i$ and $j$ be $\mathbf{w}_i$ and $\mathbf{w}_j$ respectively. Namely the output layer's outputs for class $i$ and $j$ are respectively:*

$$f_i(\mathbf{x}) = \mathbf{w}_i^T H_{l-1}...H_1\mathbf{x}, \quad and \quad f_j(\mathbf{x}) = \mathbf{w}_j^T H_{l-1}...H_1\mathbf{x},$$

*where $H_i \in \mathbb{R}^{n_{i+1} \times n_i}$, and $n_1 = d$. We define two probing vectors (each of dimension $d \times 1$) for class $i$ and class $j$ as*

$$probe_i = (\mathbf{w}_i^T H_{l-1}...H_1)^T, \quad and \quad probe_j = (\mathbf{w}_j^T H_{l-1}...H_1)^T.$$

*Suppose we have the following QR decomposition:*

$$[probe_i, probe_j] = Q \begin{bmatrix} r_{11} & r_{12} \\ 0 & r_{22} \end{bmatrix},$$

*where $Q \in \mathbb{R}^{d \times 2}$. We let the projections of $\mathbf{x}_i$ and $\mathbf{x}_j$ onto the subspace spanned by the two columns of $Q$ be $\tilde{\mathbf{x}}_i$ and $\tilde{\mathbf{x}}_j$ respectively. We assume that*

$$[\tilde{\mathbf{x}}_i, \tilde{\mathbf{x}}_j] = Q \begin{bmatrix} a_{i1} & a_{j1} \\ a_{i2} & a_{j2} \end{bmatrix}.$$

*If for some input $\mathbf{x} + \Delta$, $f_j(\mathbf{x} + \Delta) > f_i(\mathbf{x} + \Delta)$, then we say that the perturbation $\Delta$ changes the label from class $i$ to class $j$. To change the predicted label from class $i$ to class $j$, we only need to add perturbation $\Delta$ to $\mathbf{x}$ on the subspace spanned by the two columns of $Q$, and the magnitude of $\Delta$ satisfies*

$$\|\Delta\| \leq \frac{|r_{11}a_{i1} - (r_{12}a_{i1} + r_{22}a_{i2})|}{\|probe_i - probe_j\|} \leq \sqrt{a_{i1}^2 + a_{i2}^2}.$$

*Proof.* Suppose $\mathbf{x} = \mathbf{x}_i$ is the ground-truth signal. We use $\mathbf{p}_i$ and $\mathbf{p}_j$ as shorts for $probe_i$ and $probe_j$. So

$$\langle \mathbf{p}_i, \mathbf{x}_i \rangle = r_{11}a_{i1}, \quad \langle \mathbf{p}_j, \mathbf{x}_i \rangle = r_{12}a_{i1} + r_{22}a_{i2}.$$

We want to add $\Delta$ to $\mathbf{x}$ such that $\langle \mathbf{p}_i, \mathbf{x}_i + \Delta \rangle < \langle \mathbf{p}_j, \mathbf{x}_i + \Delta \rangle$. Namely, $\langle \mathbf{p}_i - \mathbf{p}_j, \Delta \rangle < -\langle \mathbf{p}_i, \mathbf{x}_i \rangle + \langle \mathbf{p}_j, \mathbf{x}_i \rangle$. This is equivalent to

$$\langle \mathbf{p}_j - \mathbf{p}_i, \Delta \rangle > \langle \mathbf{p}_i, \mathbf{x}_i \rangle - \langle \mathbf{p}_j, \mathbf{x}_i \rangle = r_{11}a_{i1} - (r_{12}a_{i1} + r_{22}a_{i2}).$$

We also know that

$$\langle \mathbf{p}_j - \mathbf{p}_i, \Delta \rangle = (r_{12} - r_{11})\Delta_1 + r_{22}\Delta_2$$

So, by the Cauchy-Schwarz inequality, we can pick a $\Delta$ such that

$$\langle \mathbf{p}_j - \mathbf{p}_i, \Delta \rangle = \|\Delta\|_2 \sqrt{(r_{12} - r_{11})^2 + r_{22}^2}.$$

So there exists an arbitrarily small constant $\epsilon > 0$ and perturbation vector $\Delta$ such that

$$\|\Delta\| \leq \left| \frac{r_{11}a_{i1} - (r_{12}a_{i1} + r_{22}a_{i2})}{\sqrt{(r_{12} - r_{11})^2 + r_{22}^2}} \right| + \epsilon, \quad \text{and} \quad \langle \mathbf{p}_i, \mathbf{x}_i + \Delta \rangle < \langle \mathbf{p}_j, \mathbf{x}_i + \Delta \rangle, \qquad (7)$$

leading to a misclassified label because $f_j(\mathbf{x} + \Delta) > f_i(\mathbf{x} + \Delta)$. $\qquad\qquad\qquad\square$

As we can see from Theorem 5, one just needs to change the components of $\mathbf{x}$ in the subspace spanned by the two probing vectors. This explains the adversarial fragility of neural network based classifiers from the feature compression perspective more concretely based on the neural network architecture: one needs to only attack the compressed features used for classification decisions to fool the classifiers into making wrong decisions.

## 4 WHEN EXPONENTIALLY MANY DATA POINTS EXIST WITHIN A CLASS

In the following, we consider a case (proof provided in the appendix) where the number of data points ($2^{d-1}$) within a class is much larger than the dimension of the input data vector, and the data points of different classes are more complicatedly distributed than considered in previous theorems.

**Theorem 6.** *Consider $2^d$ data points $(\mathbf{x}_i, y_i)$, where $i = 1, 2, \cdots, 2^d$, $\mathbf{x}_i \in \mathbb{R}^d$ is the input data, and $y_i$ is the label. For each $i$, we have $\mathbf{x}_i = A\mathbf{z}_i$, where $\mathbf{z}_i$ is a $d \times 1$ vector with each of its elements being $+1$ or $-1$, and $A$ is a $d \times d$ random matrix with each element following the standard Gaussian distribution $\mathcal{N}(0, 1)$. The ground-truth label $y_i$ is $+1$ if $\mathbf{z}_i(d) = +1$ (namely $\mathbf{z}_i$'s last element is $+1$), and is $-1$ if $\mathbf{z}_i(d) = -1$. We let $C_{+1}$ denote the set of $\mathbf{x}_i$ such that the corresponding $\mathbf{z}_i(d)$ (or label) is $+1$, and let $C_{-1}$ denote the set of $\mathbf{x}_i$ such that the corresponding $\mathbf{z}_i(d)$ (or label) is $-1$.*

*Consider a two-layer neural network for classification whose hidden layer output is $\sigma(H_1\mathbf{x} + \boldsymbol{\delta_1})$, where $H_1 \in \mathbb{R}^{m \times d}$ is a random matrix with each of its elements being Gaussian, and $\boldsymbol{\delta_1}$ is the vector of bias. For each class $C_{+1}$ or $C_{-1}$, suppose that the output layer of the neural network is given by*

$$f_{+1}(\mathbf{x}) = \mathbf{w}_{+1}^T \sigma(H_1\mathbf{x} + \boldsymbol{\delta_1}) \ and \ f_{-1}(\mathbf{x}) = \mathbf{w}_{-1}^T \sigma(H_1\mathbf{x} + \boldsymbol{\delta_1}).$$

*Suppose that the hidden layer's activation function is identity, and that $H_1$ is a matrix with orthogonal columns satisfying $H_1^T H_1 = I_{m \times m}$ (for simplicity of analysis even though the results also extend to $H_1$ being general matrices, and also to multiple-layer networks with non-linear activation functions).*

*For input $\mathbf{x}_i$, suppose that the neural network satisfies*

$$f_{+1}(\mathbf{x}_i) = \begin{cases} +1, \text{if } \mathbf{z}_i(d) = +1, \\ -1, \text{if } \mathbf{z}_i(d) = -1. \end{cases}, \ and \ f_{-1}(\mathbf{x}_i) = \begin{cases} +1, \text{if } \mathbf{z}_i(d) = -1, \\ -1, \text{if } \mathbf{z}_i(d) = +1. \end{cases} \qquad (8)$$

*Let the last element of $\mathbf{z}_i$ corresponding to the ground-truth input $\mathbf{x}_i$ be denoted by 'bit'. Then*

- *with high probability, there exists a constant $\alpha > 0$ such that the smallest distance between any two data points in the different classes is at least $\alpha\sqrt{d}$, namely $\min_{\mathbf{x}_i \in C_{+1}, \ \mathbf{x}_j \in C_{-1}} \|\mathbf{x}_i - \mathbf{x}_j\|_2 \geq \alpha\sqrt{d}$.*

- *Given a data $\mathbf{x} = \mathbf{x}_i$, with high probability, one can add a perturbation $\mathbf{e}$ of size $\|\mathbf{e}\|_2 \leq D$ such that $f_{-bit}(\mathbf{x}_i + \mathbf{e}) > f_{bit}(\mathbf{x}_i + \mathbf{e})$, where $D$ is a constant independent of $d$.*

As we can see in the proof, because the neural network makes classification decision based on the compressed features in the direction of the vector $Q_{:,d}$, namely the last column of $Q_1$, one can successfully attack the classifier along the directions of $Q_{:,d}$ using a much smaller magnitude of perturbation. Using the results of QR decomposition for products of Gaussian matrices in Lemma 3, the proofs of Theorem 4 and Theorem 5, we can obtain similar results in Theorem 6 for multiple-layer neural network models with general non-orthogonal weights.

## 5   WHEN THE NEURAL NETWORKS ARE GENERAL MULTIPLE-LAYER NON-LINEAR NEURAL NETWORKS

In this section, we present results showing the adversarial fragility of general non-linear multiple-layer neural network based classifiers. The results in this section show that one just needs to change the classifier's input along the direction of "compression" the classifier imposed on the input data, in order to change the outputs of the classifier towards predicting another label.

**Theorem 7.** *Consider a multi-layer neural network for classification and an arbitrary point* $\mathbf{x} \in \mathbb{R}^d$.

*From each class $i$, let the closest point in that class to $\mathbf{x}$ be denoted by $\mathbf{x} + \mathbf{x}_i$. We take $\epsilon > 0$ as a small positive number. For each class $i$, We let the the neural network based classifier's output at its output layer be $f_i(\mathbf{x})$, and we denote the gradient of $f_i(\mathbf{x})$ by $\nabla f_i(\mathbf{x})$.*

*We consider the points $\mathbf{x} + \epsilon \mathbf{x}_1$ and $\mathbf{x} + \epsilon \mathbf{x}_2$. Suppose that the input to the classifier is $\mathbf{x} + \epsilon \mathbf{x}_1$. Then we can add a perturbation $\mathbf{e}$ to $\mathbf{x} + \epsilon \mathbf{x}_1$ such that*

$$f_1(\mathbf{x} + \epsilon \mathbf{x}_1 + \mathbf{e}) = f_1(\mathbf{x} + \epsilon \mathbf{x}_2) \quad and \quad f_2(\mathbf{x} + \epsilon \mathbf{x}_1 + \mathbf{e}) = f_2(\mathbf{x} + \epsilon \mathbf{x}_2).$$

*Moreover, the magnitude of $\mathbf{e}$ satisfies*

$$\|\mathbf{e}\|_2 \leq \epsilon \|P_{\nabla f_1(\mathbf{x}), \nabla f_2(\mathbf{x})}(\mathbf{x}_1 - \mathbf{x}_2)\|_2,$$

*where $P_{\nabla f_1(\mathbf{x}), \nabla f_2(\mathbf{x})}$ is the projection onto the subspace spanned by $\nabla f_1(\mathbf{x})$ and $\nabla f_2(\mathbf{x})$.*

*If $\nabla f_1(\mathbf{x})$, $\nabla f_2(\mathbf{x})$, and $\mathbf{x}_2 - \mathbf{x}_1$ all have independent standard Gaussian random variables as their elements, changing from $\mathbf{x} + \epsilon \mathbf{x}_1$ to $\mathbf{x} + \epsilon \mathbf{x}_2$ will be $O(d)$ times more difficult (in terms of the square of the magnitude of the needed perturbation) than just changing the classifier's label locally using adversarial perturbations.*

**Remarks**: In order to make the classifier wrongly think the input is $\mathbf{x} + \epsilon x_2$ instead of the true signal $\mathbf{x} + \epsilon \mathbf{x}_1$ at the two output neurons for class 1 and 2, one just needs to add a small perturbation instead of adding a full perturbation $\epsilon(\mathbf{x}_2 - \mathbf{x}_1)$, due to compression of $\mathbf{x}_2 - \mathbf{x}_1$ along the directions of gradients $\nabla f_1(\mathbf{x})$ and $\nabla f_2(\mathbf{x})$. We can also add a small perturbation $\mathbf{e}$ to $\mathbf{x} + \epsilon \mathbf{x}_1$ such that $f_2(\mathbf{x} + \epsilon \mathbf{x}_1 + \mathbf{e}) - f_1(\mathbf{x} + \epsilon \mathbf{x}_1 + \mathbf{e}) = f_2(\mathbf{x} + \epsilon \mathbf{x}_2) - f_1(\mathbf{x} + \epsilon \mathbf{x}_2)$, with small magnitude $\|\mathbf{e}\|_2 \leq \epsilon \|P_{\nabla(f_1(\mathbf{x}) - f_2(\mathbf{x}))}(\mathbf{x}_1 - \mathbf{x}_2)\|_2$, where $P_{\nabla(f_1(\mathbf{x}) - f_2(\mathbf{x}))}$ is the projection onto the subspace spanned by $\nabla f_1(\mathbf{x}) - \nabla f_2(\mathbf{x})$.

From the proof of Theorem 7 in the appendix, we can see that in order for the neural network to have good adversarial robustness locally around $\mathbf{x}$, the direction of $\mathbf{x}_2 - \mathbf{x}_1$ should be in the span of the gradients $\nabla f_1(\mathbf{x})$ and $\nabla f_2(\mathbf{x})$. However, the subspace spanned by $\nabla f_1(\mathbf{x})$ and $\nabla f_2(\mathbf{x})$ may only contain "compressed " parts of of $\epsilon(\mathbf{x}_2 - \mathbf{x}_1)$, making it possible to use smaller-magnitude perturbation to change the classifier outputs than using a $\epsilon(\mathbf{x}_2 - \mathbf{x}_1)$ perturbation, but as effectively.

## 6   NUMERICAL RESULTS

In this section, we present our numerical results verifying theoretical predictions on adversarial fragility. In particular, we focus on the setting described in Theorem 6 (linear networks) and Theorem 7 (non-linear networks).

We first let $d$ denote the dimension of the input for the neural network. Then, for each $i$ ($i = 1, 2, \cdots, 2^d$), we have $\mathbf{x}_i = A\mathbf{z}_i$, where $\mathbf{z}_i$ is a $d \times 1$ vector with each of its elements being $+1$ or $-1$, and $A$ is a $d \times d$ random matrix with each element following the standard Gaussian distribution $\mathcal{N}(0, 1)$. The ground-truth label $y_i$ is $+1$ if $\mathbf{z}_i(d) = +1$ (namely $\mathbf{z}_i$'s last element is $+1$), and is $-1$ if $\mathbf{z}_i(d) = -1$. Then $X$ is a $d \times 2^d$ matrix where each column of $X$ represents an input data of dimension $d$.

**Linear networks**: Next, we train a linear neural network with one hidden layer for classification. The input layer of the neural network has dimension $d$, the hidden layer has 3000 neurons, and the output layer is of dimension 2. We denote the $3000 \times d$ weight matrix between the input layer and the hidden layer as $H_1$, and the weight matrix between the hidden layer and the output layer as a $2 \times 3000$ matrix $H_2$. We use identity activation function and we use the $Adam$ package in $PyTorch$

Table 1: Cosine of angles of trained models with training accuracy equal to 1, $d = 12$.

| Experiment No. | 1 | 2 | 3 | 4 | 5 | 6 | 7 | 8 | 9 | 10 |
|---|---|---|---|---|---|---|---|---|---|---|
| $\cos(\theta_1)$ | $-0.1970$ | $-0.1907$ | $-0.6017$ | $-0.2119$ | $-0.2449$ | $-0.5054$ | $-0.7794$ | $-0.5868$ | $-0.1655$ | $-0.4739$ |
| $\cos(\theta_2)$ | $-0.9992$ | $-0.9992$ | $-0.9984$ | $-0.9994$ | $-0.9955$ | $-0.9988$ | $-0.0795$ | $-0.9972$ | $-0.9993$ | $-0.9942$ |
| $\phi$ | $0.1812$ | $0.1870$ | $0.5888$ | $0.2048$ | $0.2032$ | $0.4985$ | $0.0738$ | $0.5895$ | $0.1480$ | $0.4497$ |

for training. The loss function we use in the training process is the Cross-Entropy loss function. We initialize the weights by uniform distribution [1]. The number of epochs is 20.

We consider $d = 12$. In each "run", we first randomly generate a random matrix $A$, and generate the data matrix $X$ accordingly. In generating the data matrix $X$, we multiply each of $A$'s columns by 5 except for the last column (Note that this modification will not change the theoretical predictions in Theorem 6. This is because the modification will not change the last column of matrix $R$ in the QR decomposition of $A$). Then we train a neural network as described above. We will keep the trained neural network as a valid "experiment" for study if the trained network has a training accuracy of 1. We keep generating "runs" until we have 10 valid "experiments" with training accuracy 1. Then, in Table 1, we report the results of the 10 valid "experiments" for the case $d = 12$, where the accuracy reaches 1 for each "experiment".

We let $W_1$ and $W_2$ be the first row and the second row of $W = H_2 H_1$, respectively. Note that $W_1$ and $W_2$ are just the two probing vectors mentioned in Theorem 5. For each valid "experiment", we consider two different angles, $\theta_1$ and $\theta_2$. $\theta_1$ is the angle between $W_1 - W_2$ and the last column of $A$. In terms of physical meaning, the absolute value of $\cos(\theta_1)$ means how much of the feature (the last column of $A$) is projected (or compressed) onto $W_1 - W_2$ in the neural network to make classification decisions. By similar derivation as in Theorem 5, $|\cos(\theta_1)|$ quantifies how much perturbation we can add to the input signal such that the output of the classifier is changed to the opposite label. For example, when $|\cos(\theta_1)|$ is 0.1970 in Experiment 1 of Table 1, we only need a perturbation 0.1970 of the $\ell_2$ magnitude of the last column of $A$ (perturbation is added to the input of the neural network) to change the output of this neural network to the opposite label. On the other hand, the optimal decoder (the minimum distance decoder or classifier) would need the input to be changed by at least the $\ell_2$ magnitude of the last column of $A$ so that the output of the optimal decoder is changed to the opposite label.

The second angle $\theta_2$ is the angle between the first row (namely $W_1$) of $W = H_2 H_1$ and the last row of the inverse of $A$. As modeled in Theorem 6, $W_1$ should be aligned or oppositely aligned with the last row of the inverse of $A$, and thus the absolute value of $\cos(\theta_2)$ should be close to 1.

We also consider the quantity "fraction" $\phi$, which is the ratio of the absolute value of $R_{d,d}$ over the $\ell_2$ magnitude of the last column of $A$. Theorem 6 theoretically predicts that $|\cos(\theta_1)|$ (or the feature actual compression ratio) should be close to "fraction" (the theoretical feature compression ratio).

From Table 1 (except for Experiment 7), one can see that Theorem 6, the actual compression of the feature vector (the last column of the matrix $A$) onto the probing vectors ($W_1 - W_2$) and "fraction" $\phi$ (the theoretical compression ratio) accurately predict the adversarial fragility of the trained neural network for classification. For example, let us look at Experiment 9. The quantity of $\phi$ is 0.1480, and thus Theorem 6 predicts that the adversarial robustness (namely smallest magnitude of perturbation to change model's classification result) of the theoretically-assumed neural network model is only 0.1480 of the best possible adversarial robustness offered by the optimal classifier. In fact, by the actual computational trained neural network experiment, 0.1480 is indeed very close to $|\cos(\theta_1)|$=0.1665, which is the size of actual perturbation (relative to the $\ell_2$ magnitude of the last column of $A$) needed to change the practically-trained classifier's decision to the opposite label. We can also see that when the theoretically predicted compression ratio $\phi$ is small, the actual adversarial robustness quantified by $|\cos(\theta_1)|$ is also very small, experimentally validating Theorem 6's purely theoretical predictions. We also notice that $|\cos(\theta_2)|$ is very close to 1, matching the prediction of Theorem 6.

We further conduct 50 experiments and see that there are 20 experiments with training accuracy 1. Among all these 20 experiments with training accuracy 1, we noticed that there are 18 cases with

---

[1] https://community.deeplearning.ai/t/default-weight-initialization-process-in-pytorch-custom-module/436680

Table 2: Averages of cosines of angles, for $|\cos(\theta_2)| > 0.9$, $d = 12$

| Avg. of $|\cos(\theta_1)|$ | Avg. of $|\phi|$ | Avg. of $\left\|\, |\cos(\theta_1)| - |\phi| \,\right\|$ |
|---|---|---|
| 0.3645 | 0.3280 | 0.0367 |

the absolute value of $\cos(\theta_2)$ over 0.9. Furthermore, for these 18 experiments, we report 3 statistical values in Table 2.

From Table 2, we can observe that the average value of $|\cos(\theta_1)|$ is 0.3645. It means that on average, we need 0.3645 of the $\ell_2$ magnitude of the last column of $A$ be added to the input signal such that the output of the classifier is changed to the opposite label. Moreover, we can conclude from Table 2 that on average, $|\phi|$ is 0.3280. It represents that the theoretical compression ratio needed to change the classifier output is on average 0.3280. We also observe that average value of $\left\|\, |\cos(\theta_1)| - |\phi| \,\right\|$ is 0.0367, meaning the actual result is close to our theoretical analysis.

**Nonlinear networks**: We trained 1-hidden-layer ( and also multiple-hidden-layer) non-linear neural networks to test for Theorem 7. We used ReLU activation functions in the hidden layer of the neural network classifier. To generate vectors $\mathbf{x}$, $\mathbf{x}_1$ and $\mathbf{x}_2$, we first define two vectors $\mathbf{z}_{+1}$ and $\mathbf{z}_{-1}$ of dimension $d$. The first $d - 1$ elements of $\mathbf{z}_{+1}$ are the same as those of $\mathbf{z}_{-1}$, and take random values $+1$ or $-1$. The last element of $\mathbf{z}_{+1}$ is $+1$ and the last element of $\mathbf{z}_{-1}$ is $-1$. Then we define vectors $\mathbf{b}_1 = A\mathbf{z}_{+1}$, and another vector $\mathbf{b}_2 = A\mathbf{z}_{-1}$. For a stencil of 10 $\alpha$-values $[0, 0.111, 0.222, 0.333, \ldots, 0.776, 0.889, 1]$, let $\mathbf{x} = \alpha\mathbf{b}_1 + (1-\alpha)\mathbf{b}_2$ for every scalar $\alpha$. In Theorem 7, take $\mathbf{x}_1 = \mathbf{b}_1 - \mathbf{x} = (1-\alpha)\mathbf{b}_1 - (1-\alpha)\mathbf{b}_2$ and $\mathbf{x}_2 = \mathbf{b}_2 - \mathbf{x} = \alpha\mathbf{b}_2 - \alpha\mathbf{b}_1$ for every scalar $\alpha$. With $d = 12$, we calculated the projection of $\mathbf{x}_1 - \mathbf{x}_2$ onto the subspace spanned by $\nabla f_1(\mathbf{x}) - \nabla f_2(\mathbf{x})$ as $P_{\nabla f_1(\mathbf{x}) - \nabla f_2(\mathbf{x})}(\mathbf{x}_1 - \mathbf{x}_2)$. We define the following ratio $\rho = \frac{\|P_{\nabla f_1(\mathbf{x}) - \nabla f_2(\mathbf{x})}(\mathbf{x}_1 - \mathbf{x}_2)\|_2}{\|\mathbf{x}_1 - \mathbf{x}_2\|_2}$. By Theorem 7 and the discussions that follow it, we know $\rho$ is "compression rate" locally: the rate of the compression of the critical feature $\mathbf{x}_2 - \mathbf{x}_1$ onto the gradient (the feature looked at by the classifier). $\rho$ is also the ratio of tolerable worst-case perturbation of the trained neural network classifier to that of optimal classifier (locally). The smaller $\rho$ is, the less adversarially robust the trained neural network is, compared with optimal minimum-distance classifier.

For every $\alpha$, we calculate the sample mean and medians of $\rho$ over 50 accurate 1-hidden-layer non-linear neuron networks in Table 3. For example, when $\alpha = 0.444$, $\rho$ has a mean of 0.3272, meaning the trained classifier is only 0.3272 ($0.3272^2 \approx 0.10$ when considering the energy of perturbation) as adversarially robust as the optimal minimum-distance classifier. The ratios are similarly small if we train neural network classifiers with more layers. The ratios are even smaller when $d$ increases.

| $\alpha$ | 0 | 0.111 | 0.222 | 0.333 | 0.444 | 0.556 | 0.667 | 0.778 | 0.889 | 1 |
|---|---|---|---|---|---|---|---|---|---|---|
| Avg. | 0.3278 | 0.3275 | 0.3273 | 0.3270 | 0.3272 | 0.3275 | 0.3274 | 0.3281 | 0.3280 | 0.3276 |
| Medium | 0.3270 | 0.3261 | 0.3258 | 0.3255 | 0.3303 | 0.3307 | 0.3293 | 0.3322 | 0.3315 | 0.3324 |

Table 3: Averages and mediums of $\rho$

## 7 CONCLUSIONS

We study the adversarial robustness of deep neural networks for classification tasks. The adversarial robustness of a classifier is defined as the smallest possible additive perturbations that can change the classification output. We provide a matrix-theoretic explanation of the adversarial fragility of deep neural network. Our theoretical results show that neural network's adversarial robustness can degrade as the input dimension $d$ increases. Analytically we show that neural networks' adversarial robustness can be only $1/\sqrt{d}$ of the best possible adversarial robustness. Our matrix-theoretic explanation is consistent with an earlier information-theoretic feature-compression-based explanation. Limitations of this paper include the need to extend detailed theoretical analysis and numerical experiments to more general data distributions, neural network architectures, and the need to further explore the relationship between adversarial robustness and network parameters such as number of layers.

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

# A   APPENDIX

## A.1   PROOF OF LEMMA 2

*Proof.* Using the Chernoff Bound, we get that

$$P(\sum_{i=1}^{d} Z_i^2 \le d\alpha) \le \inf_{t<0} \frac{E[\Pi_i e^{tZ_i^2}]}{e^{td\alpha}}.$$

However, we know that

$$E(e^{tZ_i^2}) = \int_{-\infty}^{\infty} P(x)e^{tx^2} \, dx = \frac{1}{\sqrt{2\pi}} \int_{-\infty}^{\infty} e^{(t-\frac{1}{2})x^2} \, dx.$$

Evaluating the integral, we get

$$E(e^{tZ_i^2}) = \frac{1}{\sqrt{2\pi}} \left( \frac{2\sqrt{\pi}}{\sqrt{2-4t}} \right) = \frac{\sqrt{2}}{\sqrt{2-4t}}.$$

This gives us

$$f(t) = \frac{\Pi_i E(e^{tZ_i^2})}{e^{td\alpha}} = \left( \frac{\sqrt{2}}{e^{t\alpha}\sqrt{2-4t}} \right)^d.$$

Since $d \ge 1$ and the base is positive, minimizing $f(t)$ is equivalent to maximizing $e^{t\alpha}\sqrt{2-4t}$. Taking the derivative of this with respect to $t$, we get $e^{t\alpha} \left( \alpha\sqrt{2-4t} - \frac{2}{\sqrt{2-4t}} \right)$. Taking the derivative as 0, we get $t = \frac{\alpha-1}{2\alpha}$. Plugging this back into $f(t)$, we get

$$P(X \le d\alpha) \le \left( \alpha(e^{1-\alpha}) \right)^{\frac{d}{2}} = e^{g(\alpha)d}.$$

We now notice that the exponent $g(\alpha) = \frac{1}{2}\log(\alpha e^{1-\alpha})$ goes towards negative infinity as $\alpha \to 0$, because $\log(\alpha)$ goes to negative infinity as $\alpha \to 0$.

$\square$

## A.2   PROOF OF LEMMA 3

*Proof.* We prove this by induction over the layer index $i$. When $i = 1$, we can perform the QR decomposition of $H_1 = Q_1 R_1$, where $R_1$ is an upper triangular matrix of dimension $n_1 \times n_1$, $Q_1$ is a matrix of dimension $n_2 \times n_1$ with orthonormal columns. From random matrix theories Hassibi & Vikalo (2005); Xu et al. (2004), we know that $R_1$'s off-diagonal elements in the strictly upper triangular section are i.i.d. standard Gaussian random variables following distribution $\mathcal{N}(0,1)$.; its diagonal element in the $j$-th row is the square root of a random variable following the chi-squared distribution of degree $n_2 - j + 1$.

Let us now consider $H_2$ of dimension $n_3 \times n_2$. Then

$$H_2 H_1 = H_2 Q_1 R_1.$$

Note that $H_2 Q_1$ is a matrix of dimension $n_3 \times n_1$, and the elements of $H_2 Q_1$ are again i.i.d. random variables following the standard Gaussian distribution $\mathcal{N}(0,1)$. To see that, we first notice that because the rows of $H_2$ are independent Gaussian random variables, the rows of $H_2 Q_1$ will be mutually independent. Moreover, within each row of $H_2 Q_1$, the elements are also independent $\mathcal{N}(0,1)$ random variables because the elements are the inner products between a vector of $n_2$ independent $\mathcal{N}(0,1)$ elements and the orthonormal columns of $Q_1$. With $Q_1$ having orthogonal columns, these inner products are thus independent because they are jointly Gaussian with 0 correlation.

Then we can replace $H_2 Q_1$ with matrix $H_2'$ of dimension $n_3 \times n_1$, with elements of $H_2'$ being i.i.d. $\mathcal{N}(0,1)$ random variables. We proceed to perform QR decomposition of $H_2' = Q_2 R_2$, where $R_2$ is of dimension $n_1 \times n_1$. Again, from random matrix theories, we know that $R_2$'s off-diagonal elements in the strictly upper triangular section are i.i.d. standard Gaussian random variables following distribution $\mathcal{N}(0,1)$.; its diagonal element in the $j$-th row is the square root of a random variable following the chi-squared distribution of degree $n_3 - j + 1$.

Because

$$H_2 H_1 = Q_2 R_2 R_1,$$

and the products of upper triangular matrices are still upper triangular matrices, $Q_2(R_2 R_1)$ is the QR decomposition of $H_2 H_1$.

We assume that $H_{i+1} H_i ... H_1$ has a QR decomposition $Q_{i+1} R_{i+1} \cdots R_1$. Then by the same argument as going from $H_1$ to $H_2 H_1$, we have

$$H_{i+2} H_{i+1} H_i ... H_1 = Q_{i+2}(R_{i+2} Q_{i+1} R_{i+1} \cdots R_1)$$

working as the QR decomposition of $H_{i+2} H_{i+1} H_i ... H_1$, where $Q_{i+2}$ is an $n_{i+3} \times n_1$ matrix with orthonormal columns.

By induction over $i$, we complete the proof. $\qquad\square$

### A.3 PROOF OF THEOREM.6

*Proof.* The proof of the first claim follows the same idea as in the proof of the first claim of Theorem 1. The only major difference is that we have $2^{d-1} \times 2^{d-1} = 2^{2(d-1)}$ pairs of vectors to consider for the union bound. For each pair of vector $\mathbf{x}_i$ and $\mathbf{x}_j$, $\mathbf{x}_i - \mathbf{x}_j$ still have i.i.d. Gaussian elements with the variance of each element being at least 4. By Lemma 2 and the union bound, taking constant $\alpha$ sufficiently small, the exponential decrease (in $d$) of the probability that $\|\mathbf{x}_i - \mathbf{x}_j\|$ is smaller than $\alpha\sqrt{d}$ will overwhelm the exponential growth (in $d$) of $2^{2(d-1)}$, proving the first claim of Theorem 6. Without loss of generality, we assume that the ground-truth signal is $\mathbf{x}_i$ corresponding to label $+1$. Then we consider the QR decomposition of $H_1 A$,

$$H_1 A = Q_1 R,$$

where $Q_1 \in \mathbb{R}^{m \times d}$ satisfies $Q^T \times Q = I_{d \times d}$, and $\mathbb{R}^{d \times d}$ is an upper-triangular matrix. We further let the QR decomposition of $A$ as

$$A = Q_2 R,$$

where $Q_2 \in \mathbb{R}^{d \times d}$ and $Q_1 = H_1 Q_2$. Notice that the two QR decompositions share the same $R$ because of $H_1^T H_1 = I_{m \times m}$.

Then the weight for the class '+1' is given by $\mathbf{w}_{+1} = \frac{1}{R_{d,d}} Q_{:,d}$, and the weight for the class '-1' is given by $\mathbf{w}_{-1} = -\frac{1}{R_{d,d}} Q_{:,d}$, where $Q_{:,d}$ is last column of matrix $Q_1$. We let

$$\mathbf{e} = Q_2 \times \mathbf{e}_{basis},$$

where $\mathbf{e}_{basis} = (0, 0, ..., 0, 0, -2R_{d,d})^T$. We claim that under such a perturbation $\mathbf{e}$, the input will be $\mathbf{x}_i + \mathbf{e}$ and we have

$$f_{+1}(\mathbf{x}_i + \mathbf{e}) = -1, \text{ and } f_{-1}(\mathbf{x}_i + \mathbf{e}) = 1,$$

thus changing the classification result to the wrong label.

To see this, we first notice that the inverse of $R$ is an upper triangular matrix given by

$$\begin{bmatrix} * & * & * & \ldots & * & * & * \\ 0 & * & * & \ldots & * & * & * \\ 0 & 0 & * & \ldots & * & * & * \\ & & & \ldots & & & \\ 0 & 0 & 0 & \ldots & 0 & \frac{1}{R_{d-1,d-1}} & -\frac{R_{d-1,d}}{R_{d-1,d-1} \cdot R_{d,d}} \\ 0 & 0 & 0 & \ldots & 0 & 0 & \frac{1}{R_{d,d}} \end{bmatrix},$$

where we only explicitly write down the last two rows.

We know that $\mathbf{x}_i = A\mathbf{z}_i = Q_2 R \mathbf{z}_i$, so $\mathbf{x}_i + \mathbf{e} = Q_2(R\mathbf{z}_i + \mathbf{e}_{basis})$. Then $(f_{+1}(\mathbf{x}_i + \mathbf{e}), f_{-1}(\mathbf{x}_i + \mathbf{e}))^T$ is equal to

$$\begin{bmatrix} 0 & 0 & 0 & \ldots & 0 & +\frac{1}{R_{d,d}} \\ 0 & 0 & 0 & \ldots & 0 & -\frac{1}{R_{d,d}} \end{bmatrix} \begin{bmatrix} R_{1,d} \\ R_{2,d} \\ R_{3,d} \\ \ldots \\ R_{d-2,d} \\ R_{d-2,d} \\ R_{d,d} - 2R_{d,d} \end{bmatrix} = \begin{bmatrix} -1 \\ +1 \end{bmatrix}. \qquad (9)$$

The magnitude of this perturbation is

$$\|\mathbf{e}\|_2 = \|Q_2 \mathbf{e}_{basis}\|_2 = 2R_{d,d}. \tag{10}$$

By random matrix theory Hassibi & Vikalo (2005); Xu et al. (2004)for the QR decomposition of the Gaussian matrix $A$, we know that $R_{d,d}$ is the absolute value of a random variable following the standard Gaussian distribution $\mathcal{N}(0,1)$. Thus, there exists a constant $D$ such that, with high probability, under an error $\mathbf{e}$ with $\|\mathbf{e}\|_2 \leq D$, the predicted label of the neural network will be changed. $\qquad\square$

### A.4 Proof of Theorem 7

*Proof.* Suppose that we add a perturbation $\mathbf{q}$ to the input $\mathbf{x} + \epsilon \mathbf{x}_1$, namely the input becomes $\mathbf{x} + \epsilon \mathbf{x}_1 + \mathbf{q}$. Then

$$f_1(\mathbf{x} + \epsilon \mathbf{x}_1 + \mathbf{q}) \approx f_1(\mathbf{x} + \epsilon \mathbf{x}_1) + \nabla f_1(\mathbf{x})^T \mathbf{q} \quad \text{and} \quad f_2(\mathbf{x} + \epsilon \mathbf{x}_1 + \mathbf{q}) \approx f_2(\mathbf{x} + \epsilon \mathbf{x}_1) + \nabla f_2(\mathbf{x})^T \mathbf{q}$$

We want to pick a $\mathbf{q}$ such that

$$f_1(\mathbf{x} + \epsilon \mathbf{x}_1 + \mathbf{q}) \approx f_1(\mathbf{x} + \epsilon \mathbf{x}_2) \quad \text{and} \quad f_2(\mathbf{x} + \epsilon \mathbf{x}_1 + \mathbf{q}) \approx f_2(\mathbf{x} + \epsilon \mathbf{x}_2).$$

Apparently, we can take $\mathbf{q} = \epsilon(\mathbf{x}_2 - \mathbf{x}_1)$ to make this happen. However, we claim we can potentially take a perturbation of a much smaller size to achieve this goal. We note that

$$f_1(\mathbf{x} + \epsilon \mathbf{x}_1 + \mathbf{q}) \approx f_1(\mathbf{x}) + \epsilon \nabla f_1(\mathbf{x})^T \mathbf{x}_1 + \nabla f_1(\mathbf{x})^T \mathbf{q}$$

and

$$f_2(\mathbf{x} + \epsilon \mathbf{x}_1 + \mathbf{q}) \approx f_2(\mathbf{x}) + \epsilon \nabla f_2(\mathbf{x})^T \mathbf{x}_1 + \nabla f_2(\mathbf{x})^T \mathbf{q}.$$

We want

$$f_1(\mathbf{x}) + \epsilon \nabla f_1(\mathbf{x})^T \mathbf{x}_1 + \nabla f_1(\mathbf{x})^T \mathbf{q} = f_1(\mathbf{x}) + \epsilon \nabla f_1(\mathbf{x})^T \mathbf{x}_2,$$

and

$$f_2(\mathbf{x}) + \epsilon \nabla f_2(\mathbf{x})^T \mathbf{x}_1 + \nabla f_2(\mathbf{x})^T \mathbf{q} = f_2(\mathbf{x}) + \epsilon \nabla f_2(\mathbf{x})^T \mathbf{x}_2.$$

Namely, we want

$$\epsilon \nabla f_1(\mathbf{x})^T \mathbf{x}_1 + \nabla f_1(\mathbf{x})^T \mathbf{q} = \epsilon \nabla f_1(\mathbf{x})^T \mathbf{x}_2, \quad \text{and} \quad \epsilon \nabla f_2(\mathbf{x})^T \mathbf{x}_1 + \nabla f_2(\mathbf{x})^T \mathbf{q} = \epsilon \nabla f_2(\mathbf{x})^T \mathbf{x}_2.$$

So

$$\nabla f_1(\mathbf{x})^T \mathbf{q} = \epsilon \nabla f_1(\mathbf{x})^T (\mathbf{x}_2 - \mathbf{x}_1), \quad \text{and} \quad \nabla f_2(\mathbf{x})^T \mathbf{q} = \epsilon \nabla f_2(\mathbf{x})^T (\mathbf{x}_2 - \mathbf{x}_1).$$

Then we can just let $\mathbf{q}$ be the projection of $\epsilon(\mathbf{x}_2 - \mathbf{x}_1)$ onto the subspace spanned by $\nabla f_1(\mathbf{x})$ and $\nabla f_2(\mathbf{x})$.

If $\nabla f_1(\mathbf{x})$, $\nabla f_2(\mathbf{x})$, and $\mathbf{x}_2 - \mathbf{x}_1$ all have independent standard Gaussian random variables as their elements, then the square of the magnitude (in $\ell_2$ norm ) of that projection of $\mathbf{x}_2 - \mathbf{x}_1$ will follow a chi-squared distribution of degree 2. At the same time, the square of the magnitude of $\mathbf{x}_2 - \mathbf{x}_1$ will follow the chi-squared distribution with degree $d$. Moreover, as $d \to \infty$, the square of the magnitude of $\mathbf{x}_2 - \mathbf{x}_1$ is $\Theta(d)$ with high probability. Thus changing from $\mathbf{x} + \epsilon \mathbf{x}_1$ to $\mathbf{x} + \epsilon \mathbf{x}_2$ will be $O(d)$ times more difficult than changing the classifier's label using an adversarial attack. $\qquad\square$

