# OpenReview forum: "Towards unlocking the mystery of adversarial fragility of neural networks"
_ICLR.cc/2025/Conference — ICLR 2025 Conference Withdrawn Submission_

### Official Review · Reviewer_wdU2 · 2024-10-29

**Soundness:** 2
**Presentation:** 1
**Contribution:** 2
**Rating:** 5
**Confidence:** 2

**Summary:**

This paper studied the smallest magnitude of perturbations that could alter the model output, particularly in linear cases under several assumptions. The authors demonstrated that the adversarial robustness of models degraded as the input dimension $d$ increased. Besides, they analytically showed that the adversarial robustness of linear networks could be $1/\sqrt{d}$ of that of the minimum-distance classifier.

**Strengths:**

$\bullet$ Exploring the smallest magnitude of perturbations that can change model output is intriguing. The paper also provided detailed derivations and proofs.

$\bullet$ The comparison between the adversarial robustness of minimum-distance classifiers and DNNs is also noteworthy.

**Weaknesses:**

1. Several theorems were based on assumptions that were too strong, and provided little assurance that the analysis of "adversarial robustness of neural networks" can be generalized to any two-layer DNN, including:

$\bullet$ Theorem 1 analyzed the robustness of a two-layer linear network under the following assumptions: (a) each dimension of the training samples follows a standard Gaussian distribution, (b) the activation layer is identity matrix $I$, (c) the linear matrix $H$ is an orthogonal matrix, and (d) for the i-th sample, each i is a distinct label, with the model outputting a score of 1 for category i, and a score of 0 for all other categories, as stated in Eq (1).

$\bullet$ Theorem 4 used similar assumptions.

To improve it, the authors could include detailed discussions of how Theorems 1, 4, 6 and 7 might extend to, or provide insights into, more general neural network architectures used in practice.

2.	Writing: The authors could revise the manuscript to highlight the physical significance of each theorem, the limitations, and the insights for the DNN robustness community. For example, the authors could discuss potential implications of Theorem 7 for adversarial attacks on real-world DNNs.

$\bullet$ It is suggested to focus on the potential significance and application scenarios of each  theorem and lemma, and the general ideas of the proofs in the main text, while moving the detailed proofs (e.g. Lines 151-244) to the appendix. For example, for Theorem 7, what is the actual context in which "the classifier wrongly think the input is $x+\epsilon x_2$ instead of $x+\epsilon x_1$"?

$\bullet$ Since each theorem has different assumptions, it would be beneficial to make a table that clearly lists the assumptions of each theorem, indicating which theorems represent purely ideal cases and  which can be generalized to typical DNNs.

3.	The abstract and introduction contained overclaims. Most of the theorems presented in the main text were derived under strong assumptions, and some conclusions had its restrictions, e.g., in linear networks. For example, in the abstract, the authors demonstrated "neural network’s adversarial robustness can degrade … only $1/\sqrt{d}$ of the best possible adversarial robustness." What is "best possible adversarial robustness"? Do these results apply to any DNN, or only to linear networks, or only to two-layer linear networks? Are there some strict assumptions implicit in this conclusion? The authors should revise the abstract and introduction to clarify the conditions under which the conclusions are applicable.

Besides, the authors could conduct validation experiments on typical DNNs (e.g. validating Theorem 7 on the convolutional neural networks such as the LeNet or ResNet-18).  And a thorough discussion of the limitations is recommended.

**Questions:**

1.	Based on the assumptions in Eq. (1) and Eq. (5), under what conditions can $w_i$ satisfy these assumptions? Does this imply that new constraints have been added to $w_i$? The authors should discuss the practical feasibility of these conditions and how they relate to real-world DNNs.

2.	In Theorems 1 and 4, what does "with high probability" specifically refer to? Please provide a rigorous definition in the main text or appendix.

---

### Official Review · Reviewer_Jbfe · 2024-11-01

**Soundness:** 2
**Presentation:** 2
**Contribution:** 2
**Rating:** 3
**Confidence:** 3

**Summary:**

This paper presents a theoretical analysis of why neural networks classifiers are susceptible to adversarial perturbation, ie, adversarial fragility: specifically why small, targeted perturbations can dramatically change their classification outputs. The authors challenge existing theories, which attribute this fragility to factors like smoothness of decision functions or curvature of decision boundaries, arguing these approaches only partially address the problem. The authors present a matrix-theoretic analysis of this problem and explore how neural networks' robustness declines as the input dimension increases, theorizing that their adversarial robustness is inherently limited to approximately $1/\sqrt 𝑑$ of optimal robustness.

**Strengths:**

A better understanding of adversarial attacks and robustness of neural networks remains an important topic.

**Weaknesses:**

To the best of my knowledge, the paper's conclusion that adversarial robustness can only be $1/\sqrt d$ is already known [1, 2] and has been shown in more general settings.

- The theoretical analysis is weak, as all the theorems make important and unrealistic assumptions, i.e. normal distribution of the data, constraints on the weight matrices.
- $\ell_2$ is the only distance considered, several other papers have proposed theoretical analysis with respect to the $\ell_p$ norm (see [1, 2]).
- Theorem 1 spans 2 full pages to show a probabilistic bound over a linear network with several assumptions, but it's unclear why the authors came to all this work, since the distance to the decision boundary for a linear network can be computed in closed form.
- The paper proposes a total of 7 theorems, each of which is accompanied by a proof.
- The paper does not propose any related work
- The paper does not provide usable results
- The experimental section only proposes toy experiments

Suggestions for improving the paper:
- Instead of presenting a list of theorems, the authors should motivate their analysis and explain why it's interesting. How can these results help the community? Even if the theoretical analysis has assumptions, how can it be useful for real-world applications?
- Authors should propose a related work section and compare their analysis with other work. How is their analysis better or novel than the competing work?
- Authors should propose real-world experiments, adversarial robustness is now a mature research topic, and large-scale (e.g. ImageNet) experiments should be performed.

[1] Yang et al. Randomised smoothing of all shapes and sizes. ICML 2020
[2] Kumar et al. Curse of Dimensionality on Randomised Smoothing for Certifiable Robustness. ICML 2020

**Questions:**

See suggestions for improving the paper.

**Details Of Ethics Concerns:**

No ethical concerns.

---

### Official Review · Reviewer_wWt5 · 2024-11-03

**Soundness:** 2
**Presentation:** 3
**Contribution:** 2
**Rating:** 5
**Confidence:** 2

**Summary:**

This manuscript provides a theoretical investigation into the robustness of DNNs in classification tasks. Through rigorous matrix-theoretic analysis, they establish that the minimum adversarial perturbation—the smallest input modification required to change a network's classification decision—exhibits an intrinsic relationship with input dimensionality.

**Strengths:**

I appreciate the clear presentation and detailed theoretical derivation of this manuscript.

**Weaknesses:**

I am not an expert in this theoretical area, thus I cannot check all proof details and judge the theoretical contribution.
From my perspective, the conclusion of this work---adversarial robustness can degrade as the input dimension d increases---is not rigorous.

* What if the additional dimension of $\bf x$ is correlated with other dimensions? I.e., the new dimension does not bring any new imformation, would it degrade the robustness?
* On the other hand, if the new dimension brings new information, the new $\bf x \in R^{d+1}$ and the prior $\bf x \in R^{d}$ are drawn from different data distributions. How to compare the robustness of DNNs over different data distributions?
* How to compare the norm for variables with different dimensions? I.e., let $\bf \delta_1\in R^d$ and $\bf \delta_2\in R^{d+1}$, can we directly compare $||\delta_1||_2$ and $||\delta_2||_2$? They are in different dimensions, for example, can we say volume > area > length?

**Questions:**

ref weaknesses

---

### Official Review · Reviewer_zPdU · 2024-11-04

**Soundness:** 3
**Presentation:** 3
**Contribution:** 2
**Rating:** 5
**Confidence:** 4

**Summary:**

The paper provides a theoretical analysis of adversarial robustness in several specific contexts. It examines the dataset and adversarial perturbations across different settings, starting with a random linear network and progressing to trained multi-layer non-linear networks and arbitrary multi-class datasets. The authors conduct experiments with 12-dimensional synthetic data and linear or two-layer networks to support their theoretical findings regarding the sizes of adversarial perturbations.

**Strengths:**

Writing: The paper is well-written and effectively communicates its final goal from the outset. The intuitions behind the proofs are presented in a highly accessible manner.

Thoroughly Detailed Experiments: The experiments are described with great clarity and organization, providing all the necessary details for readers to fully understand the methodology and findings.

**Weaknesses:**

Lack of Related Research: The paper overlooks existing theoretical work on random networks, such as "Adversarial Examples in Multi-Layer Random ReLU Networks" by Bartlett et al.

Concepts: With the exception of Theorem 7, the settings discussed are largely unrelated to each other or to real-world scenarios. Theorem 7 relies heavily on linearity, (although the claim to apply on for highly non-linear networks) state only that changes in output due to input perturbations can be captured through projections on the relevant gradients.

Overstating Generality: The paper makes broad claims about phenomena related to dimensionality that are primarily observed in random networks, a point that is only briefly mentioned in the introduction and not sufficiently discussed throughout the rest of the paper.

**Questions:**

1. Doesn't the perturbation e chosen is a one gradient step attack with simple targeted loss?
2. Does  x + \epsilon x1, and x+ \epsilon x2 necessarily classified differently? it seem plausible that the classification is different only for very big \epsilon. have you tested it?
3. What does the experiments shows?

---

### Note · Authors · 2024-12-02

I have read and agree with the venue's withdrawal policy on behalf of myself and my co-authors.